# Brain State Relays Self-Processing and Heartbeat-Evoked Cortical Responses

**DOI:** 10.3390/brainsci13050832

**Published:** 2023-05-21

**Authors:** Ying Huang, Musi Xie, Yunhe Liu, Xinyu Zhang, Liubei Jiang, Han Bao, Pengmin Qin, Junrong Han

**Affiliations:** 1Key Laboratory of Brain, Cognition and Education Sciences, Ministry of Education, School of Psychology, Center for Studies of Psychological Application and Guangdong Key Laboratory of Mental Health and Cognitive Science, South China Normal University, Guangzhou 510631, China; ying_huang@m.scnu.edu.cn (Y.H.); musi_xie@m.scnu.edu.cn (M.X.); liuyunhe980525@163.com (Y.L.); zhangxinyu@m.scnu.edu.cn (X.Z.); liubei_jiang@163.com (L.J.); han_bao@m.scnu.edu.cn (H.B.); 2Pazhou Lab, Guangzhou 510330, China; 3Key Laboratory of Brain, Cognition and Education Science, Ministry of Education China, Institute for Brain Research and Rehabilitation and Guangdong Key Laboratory of Mental Health and Cognitive Science, South China Normal University, Guangzhou 510631, China

**Keywords:** brain state, self-processing, heartbeat-evoked cortical activity

## Abstract

The self has been proposed to be grounded in interoceptive processing, with heartbeat-evoked cortical activity as a neurophysiological marker of this processing. However, inconsistent findings have been reported on the relationship between heartbeat-evoked cortical responses and self-processing (including exteroceptive- and mental-self-processing). In this review, we examine previous research on the association between self-processing and heartbeat-evoked cortical responses and highlight the divergent temporal-spatial characteristics and brain regions involved. We propose that the brain state relays the interaction between self-processing and heartbeat-evoked cortical responses and thus accounts for the inconsistency. The brain state, spontaneous brain activity which highly and continuously changes in a nonrandom way, serves as the foundation upon which the brain functions and was proposed as a point in an extremely high-dimensional space. To elucidate our assumption, we provide reviews on the interactions between dimensions of brain state with both self-processing and heartbeat-evoked cortical responses. These interactions suggest the relay of self-processing and heartbeat-evoked cortical responses by brain state. Finally, we discuss possible approaches to investigate whether and how the brain state impacts the self-heart interaction.

## 1. Introduction

How is the self constructed? A three-level-self model based on a meta-analysis proposed that the self is constructed by organizing the body-environment information in a gradient way, from interoceptive-processing to exteroceptive-processing to mental-self-processing [1]. This model emphasized that interoceptive-processing served as a cornerstone for self-constructing [2,3,4,5]. Recent research has indicated that interoceptive-processing interacted intimately with both exteroceptive-processing and mental-self-processing [6]. Heartbeat-evoked cortical responses serve as a neurophysiological maker of interoceptive-processing and are widely used to investigate not only the brain–heart interaction in the past few decades but also in the interplay between the self and heartbeat. However, divergent results were found regarding the relationship between heartbeat-evoked cortical responses and self-processing at the exteroceptive and mental-self level (hereafter described as “self-processing”).

In this review, we assume that the brain state, referring to the highly variable spontaneous brain activity, relays the interaction between self-processing and heartbeat-evoked cortical responses (hereafter described as self-heart interaction), and thus contributes to the divergent results observed in such interactions. In the subsequent sections, we first scrutinize prior research to illustrate the temporal-spatial inconsistency of self-heart interaction (Section 2). We then introduce the concept and features of brain state (Section 3). Furthermore, we overview prior studies on how brain state interacts with self-processing (Section 4), and the interplay between brain state and heartbeat-evoked cortical responses (Section 5). Finally, we propose further experiments that manipulate contextual factors, such as task requirements and stimuli arrangements, to induce changes in brain state and investigate the different self-heart interactions (Section 6).

## 2. Heartbeat-Evoked Cortical Responses and Self-Processing

Heartbeat-evoked cortical responses, an index of the central monitoring of cardiac inputs [7], are frequently employed to investigate the interoceptive processing. These responses are neural activity patterns time-locked to heartbeats (e.g., commonly time-locked to peaks of R or T waves in cardiac cycles recorded in electrocardiogram) [6]. In electroencephalograms (EEG), these responses are called heartbeat-evoked potentials (HEPs) [8], while in magnetoencephalography (MEG), they are known as heartbeat-evoked responses (HERs) [6].

Heartbeat-evoked cortical responses tightly interact with the exteroceptive-processing of self, particularly in bodily self-consciousness (BSC). Three crucial aspects of BSC are self-identification with the body, self-location and the first-person perspective [9]. According to an EEG study, HEP amplitude modulation during 250–305 ms after the R peak over the frontocentral regions was found to covary with the degree of self-identification with the seen body as being their own, induced by the full-body illusions [10]. An intracranial EEG study using the same paradigm observed the self-related HEP modulation in the insula between 217 and 367 ms after the R peak [11]. Additionally, it was found that more positive HEP amplitude (195–289 ms after the R wave onset over the right central-parietal sites) corresponded to increased self-other similarity (self-identification with other’s face) induced by cardio-visual synchrony of morphing faces [12]. Furthermore, BSC is closely related to the perception of somatosensory stimuli, as it involves the subjective experience of one’s own body and its sensations [13,14]. Indeed, research found that somatosensory stimuli detection performance was associated with pre-stimulus HEP amplitudes between 296 and 400 ms over the contralateral somatosensory and central electrodes [15].

HERs have also been found to be related to mental-self-processing, including spontaneous thoughts, imagination, decision-making based on self-preference, name recognition, and emotion processing. For example, studies found that HERs amplitudes around 298–327 ms after T peaks over medial posterior sensors covaried with the self-relatedness of spontaneous thoughts [16,17]. In addition, HERs have been found to differ when participants were asked to imagine themselves versus others [18]. Self-preference was found to elicit more positive HER amplitudes at 201–262 ms after T waves across right frontal-to-parietal sites [19]. In a self/other emotion rating task, HER amplitudes during the cue processing (426–436 ms after R-peak in the right frontal region) could distinguish whether the participants were cued to adopt self or other perspective and contributed to the subjective experience of valence in self condition [20]. Our recent study further provided evidence of a dual interaction between HERs and name perception. Specifically, hearing a subject’s own name (SON) altered the HERs between 276 and 320 ms after T peaks around parietal sensors. Additionally, the SON judgement was biased by the pre-stimulus HERs between 152 and 184 ms after T peaks and prominent at the frontal and parietal sensors [21] (Figure 1).

Notably, the source reconstruction of these studies revealed divergent brain regions involved in the relationship between HEPs (or HERs) and self-processing. While some studies have identified regions in the left ventral precuneus (Pr) [16], right insula [17], anterior Pr and posterior cingulate cortex bilaterally [18] as playing a critical role, others have suggested the involvement of the right and left anterior ventromedial prefrontal cortex, right post-central complex and right supramarginal gyrus [19], right frontal operculum and occipital regions, specifically the left dorsal occipital and ventral occipital regions [20], as well as both the right and left TPJ [21]. This lack of agreement underscores the need for further research to clarify the relationship between HEPs (or HERs) and self-processing and to identify the specific cortical regions involved.

In sum, the temporal-spatial characteristics of self-related HERs (or HEPs) effect discussed above exhibit variabilities across studies. For example, the scalp distribution for BSC-related HEP effect varied across from fronto-central to parieto-central [10,11,12,15], and the effect duration was around 50–150 ms ranging from 200 to 400 ms after R peaks. Similarly, different scalp distributions were found in the self-related HERs effect at the level of mental-self, including medial posterior [16], posterior [18], right frontal-to-parietal [19], right frontal [20], around parietal [21] and prominent at the frontal and parietal sensors [21]. Additionally, the time duration of self-related HERs (or HEPs) effect was about 10–60 ms ranging from 350 to 600 ms after R peaks approximately. Additionally, the sources reconstruction of these studies showed divergent brain regions contributing to the effect. One potential explanation for this variability is that the brain state, a highly variable background of ongoing neuronal activity and tightly linked to behavioral context [22], varied across the research. For instance, the default mode network (DMN), which showed a particularly high level of activity at resting-state [23] and had a tight association with brain state [22], was found to contribute to the self-related HERs effect [16].

Thus, we assume that the brain state changing along with changes of task requirement, such as types of stimuli, probability of stimuli presentation and so on, relays the interplay between self-processing and heartbeat-evoked cortical responses (Figure 2). To converge the current evidence and to provide support for our assumption, we reviewed studies investigating the interaction between brain state and self-processing, as well as those exploring the relationship between brain state and heartbeat-evoked cortical responses. These studies are discussed in subsequent sections. 

## 3. What Is Brain State?

Brain state refers to the constantly changing, highly variable, ongoing, and spontaneous neuronal activity in the brain [22]. It reflects the multiplexing of various needs of the nervous system and is tightly linked to the behavioral context of the animal. At any given time, it can be represented as a point in a high-dimensional space, with dimensions including attention, arousal, mood, etc., where its position corresponds to the activity of relevant neural units. Additionally, over time, this ongoing neuronal activity forms a trajectory in this state space [22]. Notably, brain states change discretely or continuously. For instance, discrete brain states, such as awake and sleep states, can be formed by largely distinctive patterns of neural activity [23]. Continuous changes in brain state can be reflected by externally observable markers (e.g., pupil diameter [24] and movement [25]) or continuous internal variations in neural activity (e.g., activity variations in brain network) [26]. Additionally, variations in pre-stimulus baseline activity over time can reflect the continuous nature of brain state fluctuations. Moreover, various behavioral and neural measures, as well as self-reported measures that reflect the inner workings of thoughts and emotions, have been proposed as potential indicators of changes in brain state [22]. For instance, the level of arousal can be assessed based on heart rate [27], EEG vigilance (alpha slow-wave index) [28,29], alpha power in the occipito-parietal cortex [30,31,32,33,34], and spindles and slow oscillations measured by EEG during sleep [35]. The attentional state can manifest in measures such as reaction time [36,37,38,39,40], accuracy [40], recall percentage [41], and N2 posterior-contralateral component, which serves as a neural marker of spatial selective attention [42]. Mood states can be gauged in a self-reported way [43,44,45,46,47,48,49,50,51,52,53,54,55,56] or by physiological indices such as blood pressure [57] and cortisol levels [58].

The interaction between pre-stimulus baseline brain activity and behavioral performance provides evidence of the important role of brain state in processing external information [59,60], particularly in self-related task performance [21,61,62,63,64]. For instance, baseline brain activity in the medial thalamus and the lateral frontoparietal network has been shown to predict the subsequent perception of self-bodily stimuli [61]. High pre-stimulus alpha power has also been found to correspond with enhanced self-relatedness of pictures [62]. In a SON/UN judgement task, pre-stimulus activity levels in the right TPJ, right temporal pole, and left superior temporal gyrus were higher when SON was reported than when UN was reported [63]. Additionally, increased negative HERs preceding names bias the perception of SON [21]. Finally, during brief rest inserted in the task (6–9 s), the activity in the medial prefrontal cortex was found to predict faster reaction time for judgments of own traits compared to others’ traits [64]. These findings highlight the potential effect of brain state on self-heart interaction.

## 4. Brain State and Self-Processing

Various dimensions of brain state (e.g., arousal, attention, and mood) were found to interact with the different kinds of cognitive processing. For example, arousal changes, such as from awake to asleep, were found to be associated with auditory stimuli perception [65]. It has been proposed that functions of attention include orienting to sensory events and detecting signals for focal processing [66]. Moods were found to modulate information processing strategies, which are suggested to perform an adaptive function to respond to different environmental challenges [67,68]. The relationship between stress and cognitive function has been widely recognized [69,70]. In particular, these dimensions of brain state were suggested to be associated with self-processing. 

### 4.1. Arousal

Self-processing was suggested to require a certain level of arousal. For example, arousal changes induced by motor states manipulation (e.g., running in place, waiting in a chair, reclining in a lounge chair) were found to alter the degree of self-focus in sentence completion tasks [27]. The enhancement of a posterior positive EEG component in response to subjects’ own name (SON) during sleep stage II was found to be different from that during wakefulness, for deviance specificity of SON was absent in the former [71]. Recent research found that SON led to stronger event-related synchronization (ERS) at 2 Hz and P3 component during wakefulness than unfamiliar names. However, this effect diminished during all stages of sleep [35]. Additionally, fMRI studies showed that SON elicited stronger activity in the left auditory cortex compared to a friend’s name or an unknown name when their eyes were closed but not when their eyes were open [72]. These results suggested that the self-processing might depend on arousal level.

On the other hand, self-processing was suggested to influence the arousal level in the short term. Increased arousal level was proposed to correspond with decreased alpha power over occipito-parietal cortex during wakefulness [30,31], as demonstrated in previous studies on self-processing. For instance, the SON was shown to induce a larger decrease in alpha power over the parietal region during wakefulness compared to unfamiliar names [32]. During the face recognition task, a greater and sustained decrease in alpha-beta EEG power over the occipital area was elicited by one’s own face compared to the friend’s and other unfamiliar faces [33]. The reported reduced self-sense (sense of agency) in proportion to the increased alpha and low-beta EEG power in the parieto-occipital regions was observed [34].

### 4.2. Attention

It was suggested that self-processing might act as a global modulator of the attentional systems [36], and self-biases were affected by attention resource [37]. In the face orientation judgement task, the reaction time for own face judgments was found to be faster than for the judgments of others’ faces [38,39]. Furthermore, own face evoked a N2 posterior-contralateral (N2pc) component (a neural marker of spatial selective attention) in both unconscious and conscious conditions, whereas other faces did not [42]. Furthermore, advances in attention capturing were found to extend to self-associated stimuli, providing stronger evidence for the relationship between attention and self-processing. This is because the association between stimuli used and the self was temporally established, and thus the self-relatedness of stimuli was retained while minimizing confounding effects of the familiarity of the imperative stimulus (e.g., own face or name). In the shape-label matching task, participants responded more quickly and accurately to the shape associated with the self (e.g., “circle” was associated with “you”) than to the shape paired with the other (e.g., “triangle” was associated with “best friend”) [36,37]. It was also found that self-associated stimuli modulated the attentional breadth using a flanker task together with a shape judgement task [40]. Together, these results highlight that self-processing can modulate the attention processing. 

Moreover, the attentional bias of self-processing was suggested to be affected by attention resource [37]. A study found that the own face inside the focus of attention facilitated name judgment, but the effect diminished when the own face was outside the focus of attention [39]. Additionally, the SON presented in the periphery was less accessible for the subsequent recall task compared to when it was located within the zone of attentional focus, under high attentional load [41].

### 4.3. Mood

It has been suggested that self-processing is associated with mood, including responsivity to affect, emotional valence, and emotional arousal. When participants were exposed to a mirror to induce self-focused attention, their responsivity to affect increased, i.e., tending to report greater elation or depression after positive or negative mood induction [43]. Recent studies found a negative bias in self trait judgement, corresponding to a greater increase in self-reported anxiety and tension levels after stress induction [44]. Regarding the valence, negative mood tended to induce self-focused attention, while positive mood tended to elicit external-focused attention [45]. Additionally, negative mood induced by a reading and listening task was found to eliminate the self-prioritization in the shape-label matching task [46]. However, the valence effect on self-prioritization was not observed when considering positive mood, suggesting different mechanisms between negative and positive valence effect on the self-prioritization [47]. For emotional arousal, it was found to be positively correlated with the self-prioritization effect, that is, happiness and anxiety with higher arousal ratings corresponded to stronger effects [47]. Additionally, anxiety induced by the naturalistic stressor was found to be associated with negative bias in self trait judgement [44].

In addition, affective disorders are closely linked to the disturbance of self-processing. Features of anxiety and depression are distorted perception of self, others, and the future experienced by the individual [48]. Participants with major depressive disorder (MDD) showed a greater tendency to agree with negative self-referential statements and reject positive ones in comparison to healthy controls [49]. Additionally, the measured EEG activity of MDD patients became less negative over time for self-related decision-making conditions was found to be more negative than the non-self-related one, across frontocentral electrodes, while the healthy group did not [50]. Individuals with anxiety had a lower positive self-view compared to the control group, while those with depression exhibited a stable past-to-present self-view and an improving present-to-future self-view [51]. 

### 4.4. Summary

Self-processing was suggested to change along with the highly variable brain state. Given that the brain state underlies multiple processing processes that support the adaptation to the environment, as well as self-processing is involved in both basic and high-order processing processes [73] and thus plays a pervasive role in organizing information from the body and the environment, it is reasonable to infer that self-processing largely depends on brain state.

## 5. Brain State and Heartbeat-Evoked Cortical Responses

Convergent results suggested that viscera, especially the heart, interacted with the brain state. Visceral signals could be considered internal stimuli influencing spontaneous brain dynamics, and the continuously ascending visceral signals were monitored by the brain almost all the time [6,17,74]. In particular, heartbeat-evoked cortical responses were suggested to be related to the dimensions of brain state, such as attention, arousal, and mood.

### 5.1. Arousal

Heartbeat-evoked cortical responses were found to covary with arousal level, showing different patterns during wakefulness and during sleep. Specifically, a more positive HEP amplitude was observed when eyes were closed compared to when eyes were open [52,75]. A decrease in HEP amplitude with increased sleep depth and a following renewed increase in that during REM sleep were found in the frontocentral at about 300–400 ms after R peaks [76]. Furthermore, studies found that the HEP amplitude differed in phasic and tonic REM microstates [77]. Additionally, patients with insomnia disorder, which is related to hyperarousal, exhibited a more positive HEP amplitude compared to healthy individuals [75]. 

### 5.2. Attention

Previous studies suggested that modulation of heartbeat-evoked cortical responses might reflect a transition from external oriented attention to internal oriented attention [8,78,79]. In intracranial EEG research, counting one’s own heartbeats produced a greater HEP amplitude in a wide range of time intervals (around 100–400 ms) over the frontal regions compared to an audio recording of simulated heartbeats [80]. Additionally, a recent EEG study directly asked participants to concentrate on their own heartbeats or white noise. Results indicated that the HEP amplitude was higher for external than internal oriented attention in a time interval of 524–620 ms over central right channels [81].

### 5.3. Mood

The relationship between mood and heartbeat-evoked cortical responses has been explored in both short-term mood induction studies and in populations with affective disorders. Short-term mood induction studies found that HEP amplitude was modulated by emotional valence [82] and emotional arousal [83,84]. Stress-induced changes in cardiac output were found to be correlated with the HEP amplitude at left temporal and lateral frontal electrode locations (Pearsons *r* = −0.80) [57]. Furthermore, the infusion of cortisol (a stress hormone) was observed to lead to a higher HEP amplitude compared to a placebo infusion [58]. However, a recent study found no change in HEP amplitude after a socially evaluated cold pressor stress task, though salivary cortisol increased after the stress intervention, possibly due to the sample composition [53]. As for long-term affective disorders, the HEP amplitude of patients with major depression disorder (MDD) was reduced, compared with healthy controls in the heartbeat counting task [54]. A further study that explored the treatment’s effect on patients with MDD found a relation between the severity of MDD symptoms and the HEP amplitude, depending on treatment [55]. Apart from MDD, alterations in HERs were also identified in individuals with generalized anxiety disorder (GAD) [52] and borderline personality disorder (BPD) [56] compared to healthy controls. Specifically, for patients with GAD, a correlation was observed between the right prefrontal HEP amplitude between 240 and 460 ms after R peaks and the severity of anxiety symptoms (Pearsons *r* = 0.70) [52]. For patients with BPD, mean HEP amplitude between 455 and 595 ms was found to be negatively correlated with emotional dysregulation (Pearsons *r* = −0.30) [56].

### 5.4. Summary

The aforementioned findings demonstrated that heartbeat-evoked cortical responses varied with fluctuations in the dimensions of brain state, including states of arousal, attention and mood. Meanwhile, the brain state was suggested to serve as a background for self-processing and to be tightly linked with it (Section 4). Variations in brain state might account for the inconsistent findings on the interaction between self-processing and heartbeat-evoked cortical responses. Thus, we propose that the brain state plays a relay role in the self-heart interaction.

## 6. Future Direction 

In light of the incongruous findings concerning the self-heart interaction, we propose that this phenomenon could be mediated by changes in brain state that are related to contextual factors. To fully understand the self-heart interaction and examine this assumption, it is important to isolate the effect of brain states by manipulating contextual factors. One way to achieve this is by altering task requirements while controlling the stimuli arrangement, such as the types of stimuli used and their probability of presentation. This approach can explicitly show how task-induced changes in brain state bias the self-heart interaction. Another way is to implicitly change the brain state by adjusting the stimuli arrangement while keeping task requirements constant, such as by modulating the probability of stimuli presentation. This can demonstrate how changes in brain state implicitly modulate the self-heart interaction. Moreover, by comparing populations with vastly different baseline brain states (e.g., individuals with autism, MDD, or disorders of consciousness), we can investigate how abnormalities in brain states impact the self-heart interaction. This could potentially provide diagnostic indicators in the future. 

## 7. Conclusions

This paper proposes and validates that the brain state relays the relationship between self-processing and heartbeat-evoked cortical responses. Previous studies investigating the self-heart interaction reported inconsistent results, including differences in temporal-spatial characteristics and involved brain regions. The brain state, a highly changing and spontaneous neural activity, may account for the discrepancies observed across studies because dimensions of brain state, such as arousal, attention, and mood, were suggested to show dual interactions with both self-processing and heartbeat-evoked cortical responses. Specifically, changes in arousal level could influence self-processing, and the latter could alter the former. Changes in arousal level were also observed along with HEP (or HER) modulation. Self-processing was suggested to act as a global modulator of the attention while being affected by the attention resource, and changes in HEP (or HER) were proposed to reflect the external/internal attention shifts. Different mood states were found to correspond with distinct self-processing phenomena, and HEP (or HER) was shown to alter across different mood states. Based on the existing literature, we propose that the brain state relays the self-heart interaction. Future investigations are warranted to investigate the effect of brain state on the self-heart interaction by manipulating contextual factors.

## Figures and Tables

**Figure 1 brainsci-13-00832-f001:**
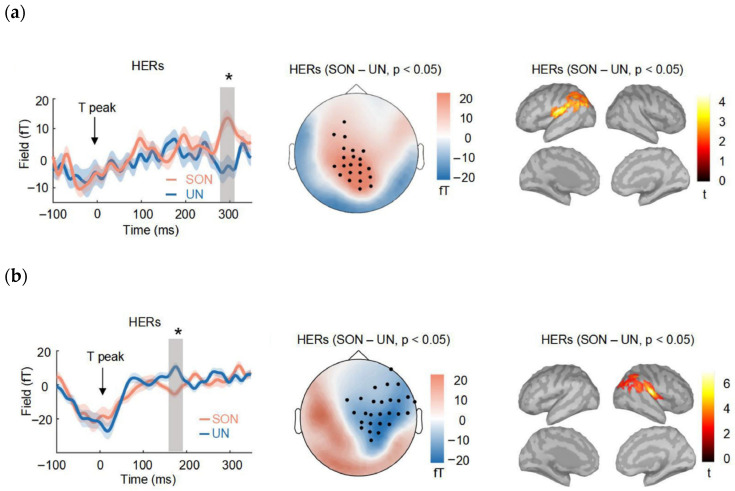
A dual interaction between HERs and name perception. (**a**) The HERs between 276 and 320 ms after T peaks around parietal sensors were different between the perception of own and unfamiliar names; (**b**) Pre-stimulus HERs between 152 and 184 ms after T peaks and prominent at the frontal and parietal sensors could predict own/unfamiliar name judgement. * *p* < 0.05. Adapted from [21]. Reproduced with permission from Zhang et al., NeuroImage; published by Elsevier, 2022.

**Figure 2 brainsci-13-00832-f002:**
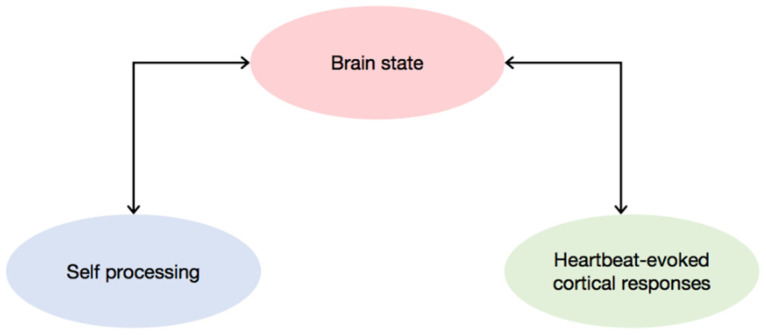
Brain state relays self-processing and heartbeat-evoked cortical responses.

## Data Availability

Not applicable.

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
