# Peer review of "Brain State Relays Self-Processing and Heartbeat-Evoked Cortical Responses"

_brainsci, 2023, doi:10.3390/brainsci13050832_

Round 1

Reviewer 1 Report

Comments and Suggestions for Authors

General Comments:

This paper presents a study on the brain-state activity and its links to self-processing and heartbeat-evoked cortical activities. Experimental results have been obtained to support these links. The study is important for research on brain-heart relationship and the proposed results are interesting. However, the paper lacks quantitative descriptions of the studied variables and relationships.

Specific Comments:

1.    The study severely lacks clear (especially, quantitative) definitions of related variables, e.g., brain state.

2.    It would be better if the discovered link between brain-state and heart-beat activities is quantitively formulated, e.g., using a correlation metric (like Pearson’s).

3.    Language Usage: Moderate Language revision is necessary. Just as examples:

(a)     Line 326: “This paper proposes the brain state relay the relationship between…” is vague and should be amended to “This paper proposes and validates a hypothesis that the brain state relays the interaction between…”.

(b)     Line 331: “because the dimensions of brain state such as ...” is vague and should be amended.

(c)     Lines 335-336: “as well as depend on attention resource” should be “while being affected by the attention resource”.

Comments on the Quality of English Language

Generally good, but moderate revision is necessary.

Reviewer 2 Report

Comments and Suggestions for Authors

The proposed manuscript is devoted to a review related to studies on the association between self-processing and heartbeat-evoked cortical responses. The authors highlight the divergent temporal-spatial characteristics and brain regions involved. They try to analyze and explain the divergent results found regarding the relationship between heartbeat-evoked cortical responses and self-processing at the exteroceptive and mental-self level.

Preliminaries to the research area are provided. In particular basic information about the models for understanding the construction of the self is provided. The role of the heartbeat-evoked cortical responses in previous investigations of the interoceptive-processing is overviewed. The impact of the brain state on self-processing and heartbeat-evoked cortical responses is analyzed.

Further experiments with contextual factors for induction of changes in brain state and investigation of different self-heart interactions are proposed.

The presentation of the main results is clear. From a formal point of view, all the contents seems to be correct. The results are valuable and worthy of being published taking into account their possible development and applications in medicine.

Minor revisions are suggested to improve the quality of the exposition:

The language should be checked and corrected.

Please state precisely what is done in every remaining section at the end of the Introduction. The structure of the manuscript should be improved. For example, there are three Sections numbered as Section 3.

Comments on the Quality of English Language

The language should be checked and corrected. Examples of grammatical mistakes are:

p. 1line 23-25: It should be ”relays” instead of ”relay”, ”accounts” instead of ”account”, and ”changes” instead of ”change”.

p. 2line 48-52: It should be ”relays” instead of ”relay”, ”contributes” instead of ”contribute”, and ” interacts” instead of ” interact”.

Round 2

Reviewer 1 Report

Comments and Suggestions for Authors

The Authors have carefully addressed the Reviewers' comments. More details have been added to clarify the presentation.